# THLANet: A deep learning framework for predicting TCR-pHLA binding in immunotherapy applications

Xu Long [1], Qiang Yang[1], Weihe Dong[1], Xiaokun Li[1,2,3,4]*, Kuanquan Wang[1], Suyu Dong[5]*, Gongning Luo[1], Xianyu Zhang[6], Tiansong Yang[7], Xin Gao[8,9,10]*, Guohua Wang[1]*

1 School of Computer Science and Technology, Harbin Institute of Technology, Harbin, China, 2 School of Computer Science and Technology, Heilongjiang University, Harbin, China, 3 Postdoctoral Program of Heilongjiang Hengxun Technology Co., Ltd., Harbin, China, 4 Shandong Hengxun Technology Co., Ltd., Qingdao, China, 5 College of Computer and Control Engineering, Northeast Forestry University, Harbin, China, 6 Department of Breast Surgery, Harbin Medical University Cancer Hospital, Harbin, China, 7 Department of Rehabilitation, The First Affiliated Hospital of Heilongjiang University of Traditional Chinese Medicine, Harbin, China, 8 Computer Science Program, King Abdullah University of Science and Technology (KAUST), Thuwal, Kingdom of Saudi Arabia, 9 Center of Excellence for Smart Health (KCSH), King Abdullah University of Science and Technology (KAUST), Thuwal, Kingdom of Saudi Arabia, 10 Center of Excellence for Smart Health (KCSH), King Abdullah University of Science and Technology (KAUST), Thuwal, Kingdom of Saudi Arabia

* 24B303024@stu.hit.edu.cn (XL); dongsuyu@126.com (SD); xin.gao@kaust.edu.sa (XG); ghwang@hit.edu.cn (GW)

**Data availability statement:** THLANet is available on github (https://github.com/ChanganMakeYi/THLAnet), HLA allele immunogenicity data were obtained

## Abstract

Adaptive immunity is a targeted immune response that enables the body to identify and eliminate foreign pathogens, playing a critical role in the anti-tumor immune response. Tumor cell expression of antigens forms the foundation for inducing this adaptive response. However, the human leukocyte antigens (HLA)-restricted recognition of antigens by T-cell receptors (TCR) limits their ability to detect all neoantigens, with only a small subset capable of activating T-cells. Accurately predicting neoantigen binding to TCR is, therefore, crucial for assessing their immunogenic potential in clinical settings. We present THLANet, a deep learning model designed to predict the binding specificity of TCR to neoantigens presented by class I HLAs. THLANet employs evolutionary scale modeling-2 (ESM-2), replacing the traditional embedding methods to enhance sequence feature representation. Using scTCR-seq data, we obtained the TCR immune repertoire and constructed a TCR-pHLA binding database to validate THLANet's clinical potential. The model's performance was further evaluated using clinical cancer data across various cancer types. Additionally, by analyzing divided complementarity-determining region (CDR3) sequences and simulating alanine scanning of antigen sequences, we provided new insights into the 3D binding interactions of TCRs and antigens. Predicting TCR-neoantigen pairing remains a significant challenge in immunology, THLANet provides accurate predictions using only the TCR sequence (CDR3$\beta$), antigen sequence, and class I HLA, offering novel insights into TCR-antigen interactions.

from the IEDB database (https://www.iedb.org), VDJdb (https://vdjdb.cdr3.net/) and McPAS-TCR (http://friedmanlab.weizmann.ac.il/McPAS-TCR/). The detailed information of the 10x Genomics cohort is available at: (https://www.10xgenomics.com/datasets). All relevant data are within the manuscript and its Supporting information files.

**Funding:** This work was supported by the Natural Science Foundation of Heilongjiang Province of China (ZD2024F001 to GW), National Natural Science Foundation of China (62225109 to GW, 62450122 to GW, 62372135 to GL, 62202092 to SD), and the King Abdullah University of Science and Technology (KAUST) Office of Research Administration (ORA) under Award No REI/1/5234-01-01, REI/1/5414-01-01, REI/1/5289-01-01, REI/1/5404-01-01, REI/1/5992-01-01, URF/1/4663-01-01, Center of Excellence for Smart Health (KCSH) under award number 5932, and Center of Excellence on Generative AI under award number 5940 to XG. The funders did not play any role in the study design, data collection and analysis, decision to publish, or preparation of the manuscript.

**Competing interests:** The authors have declared that no competing interests exist.

## Author summary

T-cell receptor (TCR) recognition of peptide-human leukocyte antigen (pHLA) complexes is fundamental to immune responses. However, predicting their binding poses a significant challenge due to the intricate dynamics of their interactions. We developed THLANet, a novel deep learning model, to address this challenge by integrating the ESM-2 and Transformer-Encoder modules. This approach enhances sequence feature encoding, improving the model's generalization capability and enabling accurate predictions of TCR-pHLA binding in clinical datasets. Using data processed from open-source databases, THLANet outperformed existing methods, such as PanPep and pMTnet, in precision-recall metrics across multiple epitopes. Additionally, THLANet demonstrates superior capability in identifying critical binding sites within 3D structures, providing structural insights into TCR-pHLA interactions. THLANet offers a robust framework for advancing immunotherapy research, with potential applications in the development of personalized medicine.

## Introduction

Cancer arises from genetic aberrations, including small variants like single-nucleotide substitutions, insertions, and deletions [1–3]. These alterations are unique to cancer cells and absent in normal tissues [4,5]. As a result, protein products derived from these mutations hold the potential to deliver clinical benefits without inducing toxicity in healthy tissues [6–8]. The development of strategies to harness the host immune system against malignant tumor cells marks a significant paradigm shift in cancer immunotherapy. [9–12]. Recently, immunotherapies such as adoptive cell therapy (ACT) [13,14], Immune checkpoint blockade, and tumor antigen vaccines have achieved significant clinical success [13,20,21]. The therapeutic efficacy largely stems from the anti-tumor activity of CD8+ T-cells, which target malignant cells by recognizing tumor-associated neoantigens and tumor-specific antigens on their surface (As shown in Fig 1a) [15–17]. Consequently, accurately evaluating a T-cell repertoire's capacity to interact with tumor antigens is essential for identifying and targeting cancer cells and remains central to optimizing cancer immunotherapy [18,19,22].

Human leukocyte antigen (HLA) molecules present neoantigenic peptides that are recognized by T-cell receptors (TCRs), triggering the transformation of naive T-cells into CD8+ cytotoxic T-cells. This activation stimulates the immune system, enabling the targeted destruction of malignant cells (Fig 1a) [23]. Each T-cell clone possesses a unique TCR, which can act as an antigenic peptide to safeguard the immune system against malignancies [24]. The complementarity-determining region 3 (CDR3) of the TCR $\beta$-chain is particularly significant due to its highly diverse antigen specificity, making it the focal point of our study on TCR-neoantigen interactions. Since TCRs recognize neoantigens under HLA constraints, involving both antigenic peptides and polymorphisms, understanding the TCR ternary complex's binding mechanism is crucial. The ternary complex, comprising the TCR, antigenic peptide, and HLA (TCR-pHLA), is crucial for autoimmune antigen discovery and cancer vaccine development. Advances in artificial intelligence computational models have enabled precise prediction and identification of TCR-pHLA pairings, paving the way for groundbreaking research in modern immunology [25].

Several computational tools are available to analyze TCR models and predict antigenic peptide-TCR binding specificity. Previous studies have classified these tools into three categories: (1) tools for defining TCR clusters and deciphering antigen-specific binding models,

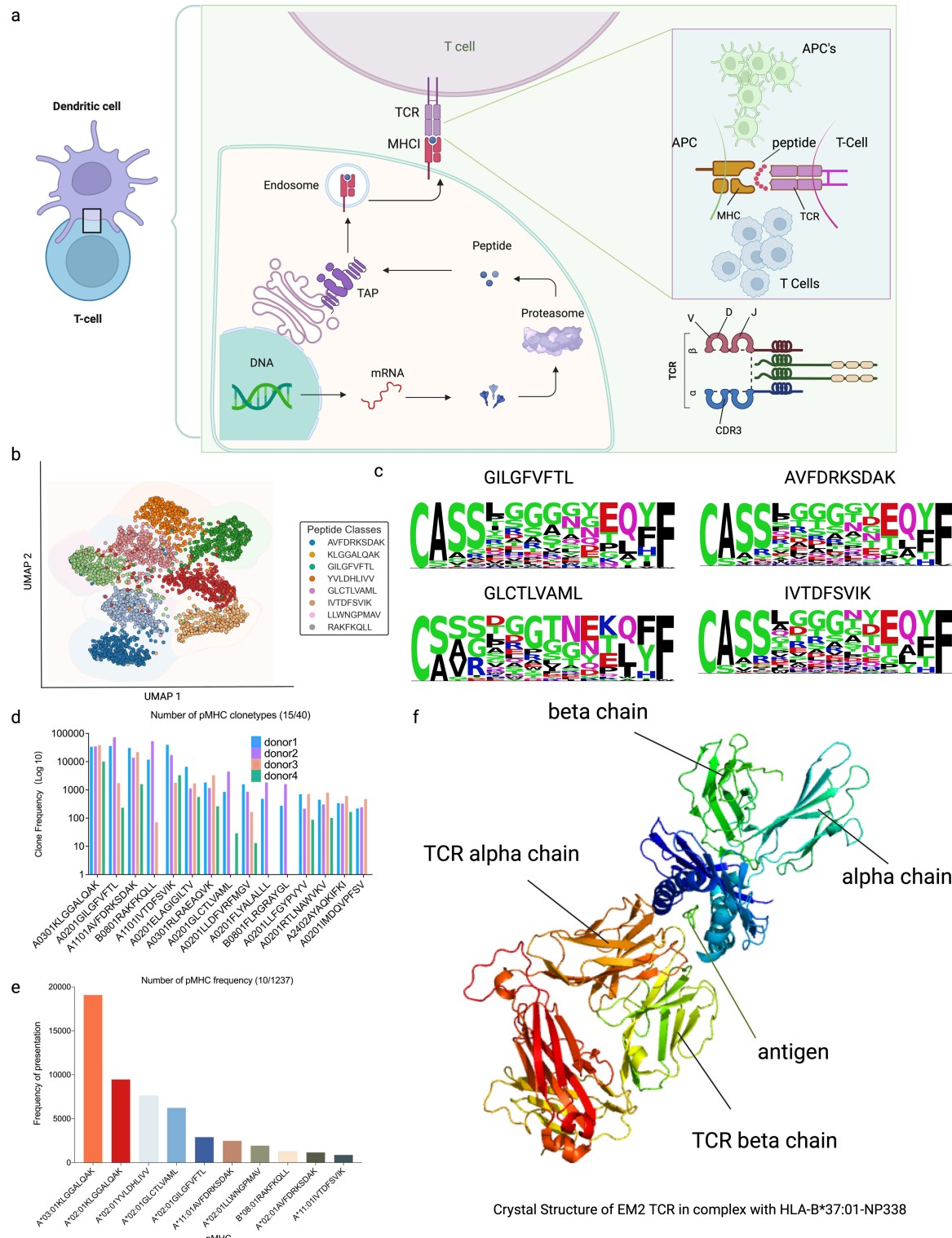

**Fig 1. Statistical information on immunogenicity data.** (**a**) Schematic diagram of TCR–peptide-major histocompatibility complex (pMHC) recognition by CD8+ T-cells. (**b**) Uniform manifold approximation and projection (UMAP) projections of the predicted epitope-specific TCR clusters. (**c**) Sequence motifs of CDR3$\beta$ representing the epitope-specific TCRs for the four protein epitopes. (**d**) Distribution of pMHC clonality in data from four donors using 10X Genomics. (**e**) Distribution of pMHC data from the VDJ database (VDJdb) and immune epitope database (IEDB). (**f**) 3D binding schematic of TCR-pHLA (from PDB ID: 6MTM). Created with BioRender.

such as TCRdist [26], DeepTCR [27], GIANA [28], and GLIPH2 [29]; (2) tools for predicting antigen-peptide specific TCR binding models, including T-cell receptor gaussian process (TCRGP) [30], TCRex [31], and NetTCR-2.0 [32]; and (3) tools for developing pan-peptide TCR binding prediction models trained on known bound TCRs, such as pMTnet [33], PanPep [34], TABR-BERT [35], DLpTCR [36], and TITAN [37]. However, the first and second categories lack the robustness needed to address the broad-spectrum TCR-pHLA recognition problem. Furthermore, models in the third category have not sufficiently explored the TCR-pHLA binding mechanism.

We propose THLANet, a deep learning framework leveraging evolutionary scale modeling (ESM-2) embeddings to predict the binding specificity between class I pHLA and TCR [38]. The architecture of THLANet as shown Fig 2. THLANet advances TCR–pHLA binding prediction by integrating ESM-2 with a bilinear attention network and segment-based CDR3 analysis, offering predictive structural insights and enhanced generalizability compared to existing methods like PanPep [34], pMTnet [33], and TABR-BERT [35]. Its novel architecture and application to immunotherapy design address critical gaps in clinical translation. To validate the model, we incorporated multiple independent datasets, including THLANet's assessment of the TCR-pHLA 3D binding conformation, providing insights into the 3D binding mechanism [39,40,42,43]. Additionally, we compiled cancer-related TCR-pHLA binding data from various peer-reviewed publications and applied THLANet for predictions, demonstrating its potential for clinical applications in cancers [44,45].

## Materials and methods

### Data curation

In this study, we curated multiple datasets to support our analysis, including a foundational dataset, a 3D crystal structure dataset of TCR-pHLA complexes, and a dataset of tumor-mutant immunogenic antigens.

We began by analyzing data generated using the 10x Genomics Chromium single-cell immunoassay platform, which leverages feature barcoding technology to create single-cell 5' libraries and V(D)J-enriched TCR sequence libraries. This platform also employs highly multiplexed pHLA multimers to identify binding specificity. Specifically, we examined four single-cell datasets obtained from the peripheral blood of four healthy donors with no known viral infections (The details of this dataset can be found in Sect E of S1 Text. These datasets included 44 CD8+ T-cell profiles that demonstrated specific binding to pHLA complexes (as shown in S5 Data). We investigated the clonal expansions of T cells to identify TCRs capable of interacting with pMHC complexes, quantifying them using original unique molecular identifiers (UMIs). When the UMI count reached or exceeded 10, the pMHC-TCR pair was deemed capable of interacting with T cells and was classified as a positive sample. From these single-cell immunoassay datasets, we compiled TCR-pHLA datasets containing 19,657 entries.

To ensure data quality, we processed the data using the open-source databases VDJdb (N = 36,168) and the IEDB (N = 31,836), resulting in high-confidence datasets [39,40]. The peptides in these databases include a mix of clinically relevant epitopes (e.g., tumor-associated neoantigens from cancers like melanoma and gastrointestinal tumors) and non-clinical or viral epitopes that serve as proxies for validating the model's binding prediction capability. While some peptides have direct clinical relevance in immunotherapy applications, such as neoantigens derived from somatic mutations in cancer patients, others (e.g., viral peptides) are used primarily to train and evaluate the model's generalizability in predicting TCR-pHLA interactions, without implying direct therapeutic use. Given the critical role of the CDR3

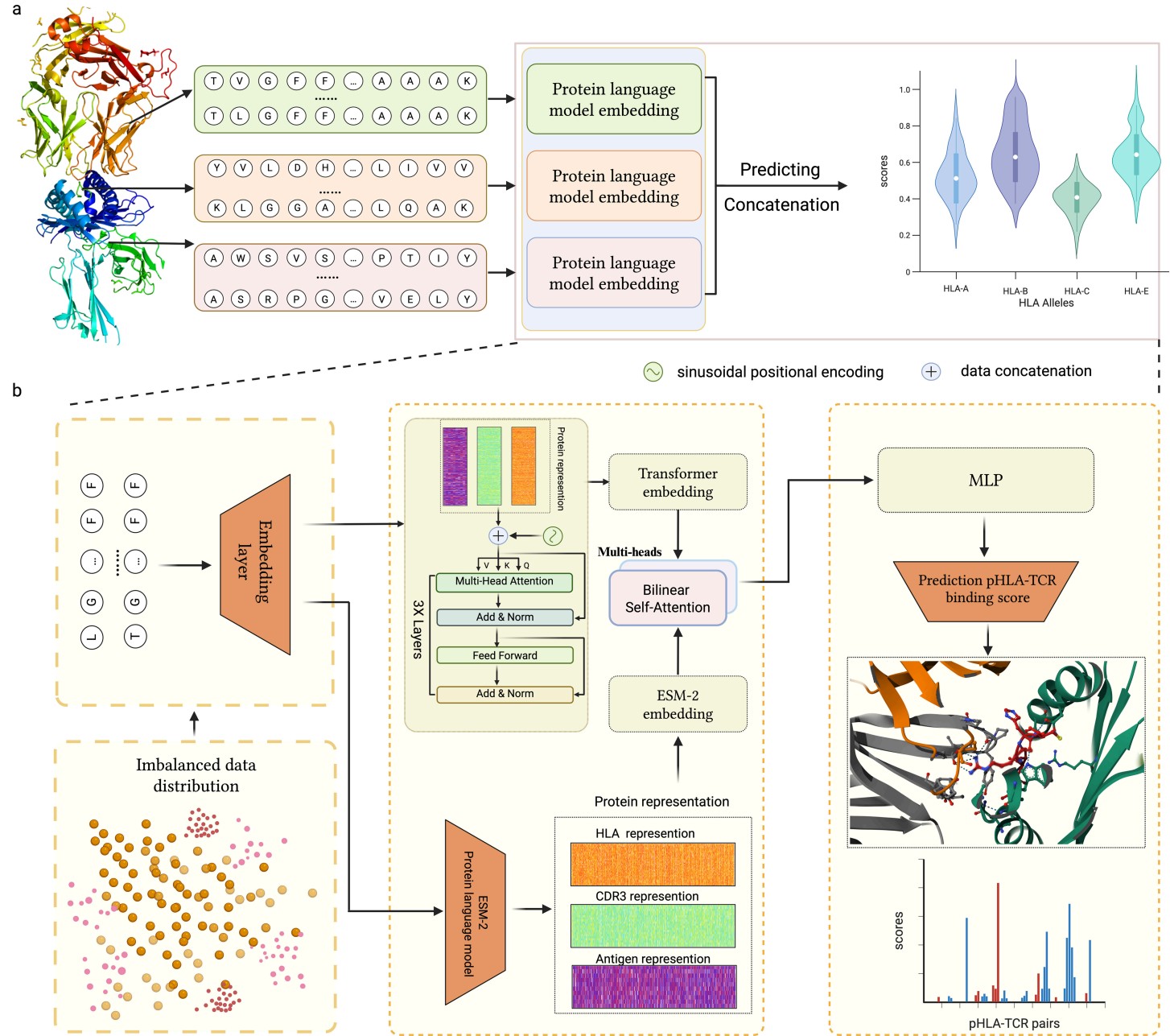

**Fig 2. Workflow of THLANet for predicting T-cell receptor (TCR)-peptide-human leukocyte antigen (pHLA) interactions.** (**a**) The pipeline of THLANet for predicting the TCR-pHLA triad group interaction process. (**b**) Detailed architecture of THLANet: The training data exhibit a long-tail distribution. Protein sequences are processed through the ESM2 module and a convolutional neural network (CNN) to capture long-distance dependencies and features. Concurrently, the sequences are initially encoded using the BLOSUM62 matrix and embedded through a transformer encoder. The two feature matrices are fused via a bilinear attention network. In the prediction module, a multilayer perceptron (MLP)-based model predicts the interaction scores between TCR and pHLA. Created with BioRender.

region of the TCR beta chain in epitope recognition, we focused exclusively on recording the CDR3 beta chain. The integrated dataset (N = 87,661) was further refined by removing duplicates and dichotomous samples, yielding a final dataset of 66,314 unique entries. We retained laboratory-validated negative samples from the IEDB database and selected samples with a

confidence score of 0 from the VDJdb database, collectively constructing the negative dataset for our study.

TCRs are highly promiscuous, recognizing millions of peptides, risking false negatives in TCR-pHLA datasets. Random mismatching in some models ignores this, causing errors. We used laboratory-validated data from 10x Genomics, VDJdb, and IEDB, ensuring accurate positive and negative samples. This minimized false negatives, enhancing THLANet's training reliability and improving its predictive accuracy for immunotherapy.

As shown in S3 Fig and Fig 1d , we analyzed cell type distributions across the four single-cell sequencing datasets and examined the clonotype distribution of pHLA. A total of 40 pHLA sequence datasets demonstrated significant clonal expansions (>10 clones) in donor T-cells. We visualized the clonal distribution of the top 15 pHLAs in CD8+ T-cells. Additionally, we cataloged 1,237 distinct pHLA types across all datasets, evaluated their overall frequencies, and highlighted the top 10 pHLA types by occurrence (Fig 1e). The pronounced coupling of certain pHLA types underscores their frequent occurrence in clinical settings, validating the clinical relevance of our findings.

To rigorously and comprehensively evaluate THLANet, we constructed a baseline dataset derived from the McPAS-TCR [41], IEDB and VDJdb databases. This dataset excluded the 10x Genomics data from the original dataset and incorporated an additional 2,904 validated samples from McPAS-TCR, resulting in 40,748 unique TCR-pHLA pairs. This diverse dataset strengthens THLANet's evaluation across varied immunological contexts, facilitating its application in personalized immunotherapy.

For the TCR-pHLA crystal structure data, we curated 3D structural information from screenings in the IEDB and retrieved 198 crystal structure datasets from the PDB database using their ID information [42,43]. After excluding low-quality entries, we finalized a dataset of 112 high-quality TCR-pHLA crystal structures (S2 Data). The spatial configuration of the pHLA–TCR triad is illustrated in Fig 1f (derived from PDB ID: 6MTM).

We also explored the physiological significance of TCR-pHLA interactions characterized by THLANet, focusing on tumor neoantigens arising from somatic mutations. To this end, we compiled TCR-pHLA pairs from peer-reviewed studies involving tumor-infiltrating lymphocytes derived from melanoma and gastrointestinal cancers [44,45]. These pairs were identified through high-throughput immunology screenings, enabling the characterization of TCR-pHLA interactions in these two highly immunogenic tumor types. These cancer-derived peptides have direct clinical relevance, as they represent tumor-specific neoantigens that could inform immunotherapy strategies, while serving as validation proxies for THLANet's binding predictions in real-world applications. This analysis offers clinically relevant insights into the mechanisms of tumor antigen presentation within the tumor microenvironment.

## Data embedding

We employed evolutionary scale modeling 2 (ESM-2) to encode amino acid sequences for protein analysis [38]. ESM-2, an unsupervised transformer-based language model, leverages an attention mechanism to identify interaction patterns between amino acid pairs in input sequences. It plays a critical role in ESMFold's success in predicting protein structures. The deep features embedded by ESM-2 capture essential three-dimensional conformational information, providing theoretical insights into TCR-pHLA binding mechanisms in three-dimensional space. As a pretrained model, ESM-2 leverages extensive prior protein knowledge, which may increase the risk of overfitting. To adapt ESM-2 to our specific encoding scenario, we fine-tuned it using HLA, antigen, and TCR sequence data, enhancing its relevance to TCR-pHLA interactions, For specific parameters of the fine-tuning process,

refer to S6 Table. To further mitigate overfitting and enrich encoded features, we processed protein sequences using the BLOSUM62 matrix (as shown in sub-figure **a** of S1 Fig) and applied sinusoidal encoding to improve spatial sequence representation [46,47]. Sinusoidal position coding enables the model to capture relative distances and sequence information between amino acid positions, guiding attention based on positional context. For unified encoding with ESM-2, we utilized a transformer encoder architecture to effectively embed the data. To seamlessly integrate the datasets generated by ESM-2 and the transformer encoder while minimizing information loss, we implemented a bilinear attention network. This approach efficiently fuses the embeddings, preserving rich joint features for a comprehensive and accurate representation.

We selected the ESM-2 650M parameter version (33 layers, 650M parameters) for TCR-pHLA sequence encoding due to its optimal balance of feature representation depth and computational efficiency. Compared to the 150M model, the 650M version offers richer sequence feature representations, improving the modeling of complex TCR-pHLA interactions. Compared to the 15B model, its lower memory requirements better suit our hardware. Pretraining on the UniRef dataset equips ESM-2 with robust prior protein knowledge, ideal for capturing three-dimensional conformational features. Experiments confirmed its efficacy, with superior the Area Under the Receiver Operating Characteristic Curve (ROC-AUC) and the Area Under the Precision-Recall Curve (PR-AUC) scores on our dataset (Fig 3), validating its suitability.

As shown in Fig 1b, we conducted a two-dimensional uniform manifold approximation and projection (UMAP) analysis of antigen-associated samples processed by the bilinear attention network. The samples, representing the top eight antigens by frequency in the training dataset, were visualized to evaluate the encoded feature distribution. The visualization clearly shows distinct distribution patterns, with samples grouped by antigen type exhibiting well-defined regional clustering in two-dimensional space.

## Bilinear attention network

The bilinear attention network is an attention-based model designed primarily for multimodal learning tasks, such as visual question answering (VQA) [48]. In VQA, given an image and a natural language question, the system generates an answer by matching text and image data. The bilinear attention network excels in preserving rich joint information while maintaining computational efficiency, offering a promising approach for enabling interaction between ESM-2 and transformer encoder data in our study [49].

## Pseudosequence construction and data preparation

As shown in sub-figure **b** of S1 Fig, the pseudosequence of the HLA sequence is constructed by selecting 34 amino acid residues to characterize the polymorphism of HLA-A, -B, and -C alleles, with these residues identified as the positions within the binding groove most likely to directly interact with the antigenic peptide, thereby effectively capturing the interaction characteristics between the antigenic peptide and the HLA molecule. The selection of these 34 amino acids is based on their critical role in various binding conformations, ensuring a robust representation of HLA-peptide interactions [50]. To prepare the data for modeling, the input sequences are aligned by standardizing the lengths of antigenic residues and CDR3$\beta$ to 15 and 25, respectively, with shorter sequences padded using the placeholder <pad> to ensure uniformity. Additionally, the original dataset is partitioned into training, validation, and test sets in an 8:1:1 ratio, respectively, to facilitate subsequent model training and evaluation.

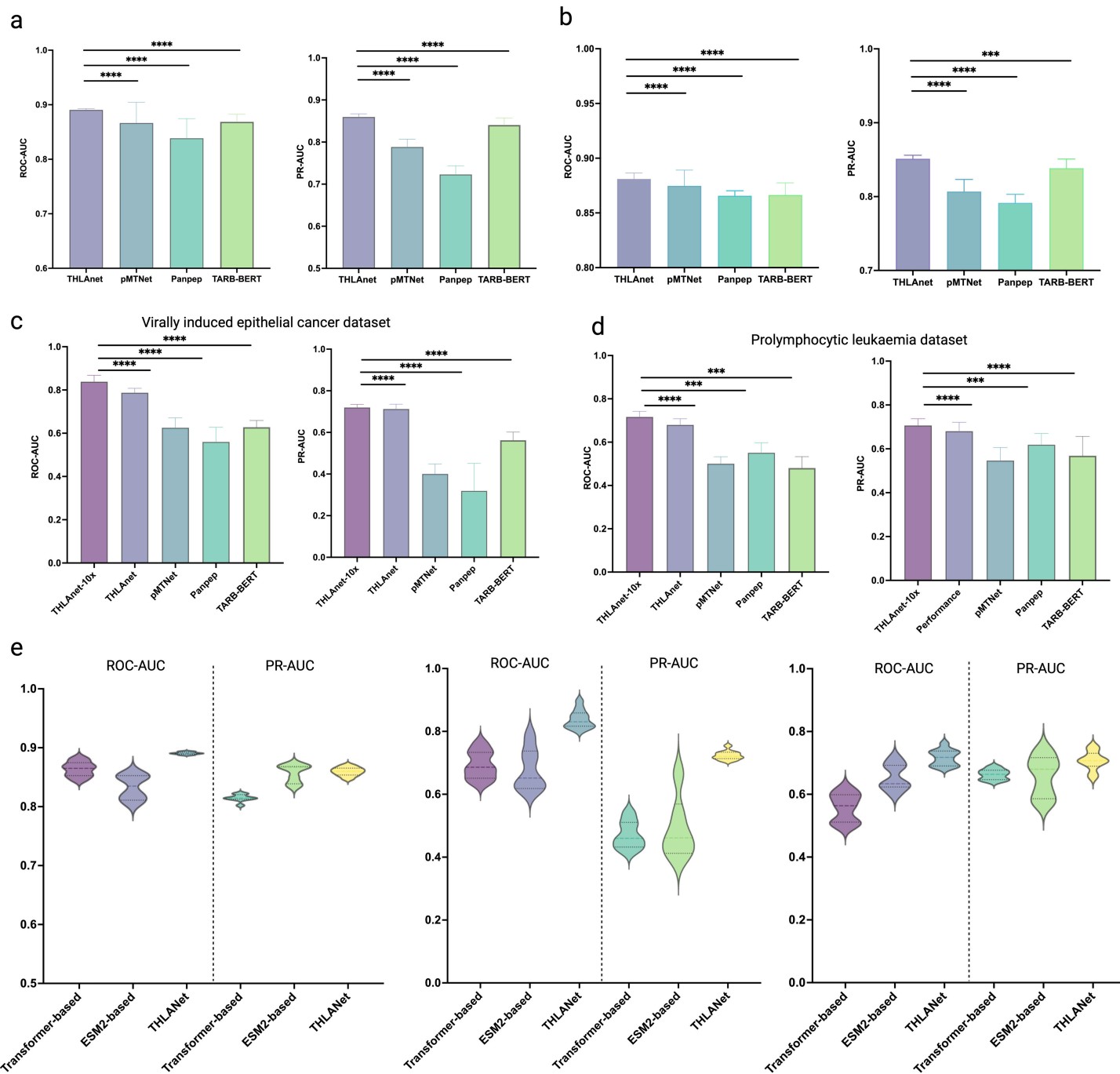

**Fig 3. THLANet prediction results.** (**a**) ROC-AUC and PR-AUC in the test dataset including 10X Genomics data. (**b**) ROC-AUC and PR-AUC in the test dataset from public databases. (**c**) ROC-AUC and PR-AUC in melanoma and gastrointestinal cancer datasets. (**d**) ROC-AUC and PR-AUC in prolymphocytic leukemia dataset. (**e**) The ROC-AUC and PR-AUC in ablation experiment.

## THLANet architecture

**ESM-2 architecture.** ESM-2, the most advanced protein language model to date, incorporates significant advancements in architecture, training parameters, computational resources, and

data scale. Trained on protein sequences from the UniRef database, ESM-2 employs a masked prediction approach where 15% of amino acids are randomly masked, and the model is tasked with predicting these positions. This training objective emphasizes amino acid prediction while requiring the model to capture complex internal representations of input sequences to achieve high accuracy. ESM-2 encodes each protein sequence into a 1280-dimensional feature vector, enabling detailed exploration of protein sequence characteristics and serving as a robust tool for protein research. For input processing, ESM-2 prepends a <cls> token at the beginning of each sequence to mark its start and appends an <eos> token at the end to denote its conclusion. When applied to encode antigenic peptides, HLA sequences, and CDR3$\beta$ regions, the resulting feature vectors are concatenated to form a unified matrix with the dimensions [n, 80, 1280], where n represents the number of samples, 80 corresponds to the total length of the concatenated sequence (including the standardized lengths and special tokens), and 1280 is the dimensionality of the feature vector for each position.

**Transformer-encoder architecture.** To enhance the feature richness of the prediction samples, we employed the Transformer-Encoder module to generate a novel encoding matrix. Each pHLA-TCR pair was encoded into a 80×21 matrix, referred to as the TCR-pHLA matrix, using the BLOSUM62 matrix. To incorporate positional information, sinusoidal positional encoding was applied to each amino acid within the matrix. This encoding method enables the model to capture the relative distances and sequential relationships between amino acid positions within the protein, thereby guiding the model's attention mechanism more effectively based on positional context. The resulting TCR-pHLA matrix, now augmented with positional information, was utilized as the input to the Transformer encoder. After processing through three layers of the Transformer encoder, the final encoded matrix was obtained, in which each amino acid is enriched with global contextual information derived from the entire sequence.

**THLANet prediction model.** The THLANet prediction model integrates of CNN layers with MLP layers (the architecture as shown in S4 Fig). The CNN (TextCNN) employs three 1D convolutional layers with kernel sizes of [1, 2, 1] and 1000 filters to extract features from text sequences. ReLU activation and max pooling operations follow the convolutional layers, and the pooled outputs are concatenated into a feature vector. This vector passes through three fully connected layers with 1048 and 512 units, each activated by ReLU, followed by batch normalization layers to stabilize training. A dropout layer (rate = 0.4) mitigates overfitting. Finally, a softmax layer outputs a binary classification probability distribution. Model optimization utilized the Adam optimizer with a learning rate of $5 * 10^{-4}$ and a batch size of 128. Training ran for up to 100 epochs, with early stopping triggered if validation loss failed to decrease for five consecutive epochs.

## CDR$\beta$ has binding preference

We analyzed the amino acid preferences of CDR3$\beta$ sequences associated with responses to eight distinct protein epitopes: AVFDRKSDAK, KLGGALQAK, GILGFVFTL, YVLDHLIVV, GLCTLVAML, IVTDFSVIK, LLWNGPMAV, and RAKFKQLL (Fig 1c and S2 Fig). While the N- and C-terminal regions of the epitope-specific CDR3$\alpha\beta$ loops exhibited relative conservation, the regions directly interacting with the epitopes showed substantial diversity. Interestingly, specific amino acids such as glycine (G), leucine (L), proline (P), and polar residues like asparagine (N), serine (S), threonine (T), and tyrosine (Y) were more frequently identified in the core positions of CDR3$\beta$ sequences.

### Dynamic sampling strategy

To address the long-tail distribution and data dependency in TCR-pHLA interaction datasets, we implemented an HLA allele-specific sampling strategy during THLANet's training. This approach dynamically weighted samples based on HLA allele frequency to balance rare and common alleles. By mitigating bias from overrepresented alleles, it enhanced model robustness and generalization.

### Experimental setting

**THLANet training.** THLANet was implemented using Python 3.10, PyTorch 2.5.0, and Compute Unified Device Architecture (CUDA) 12.4. The computational environment included an Intel(R) Xeon(R) Gold 6248R CPU @ 3.00GHz with 256 GB of RAM and two NVIDIA A100-PCIE-40GB GPUs, each offering 40 GB of memory. The model achieved complete convergence after 60 epochs. To optimize THLANet's performance, we conducted grid search to tune hyperparameters, including optimizer (Adam, SGD), learning rate $[1 * 10^{-4}, 5 * 10^{-4}, 1 * 10^{-3}]$, batch size [64, 128, 256], and weight decay $[1 * 10^{-5}, 1 * 10^{-4}]$). Using ROC-AUC as the primary metric on the validation set, we identified Adam (for adaptive gradient adjustment), a learning rate of $5 * 10^{-4}$, and a batch size of 128 as optimal. Combined with early stopping (patience 5), these hyperparameters yielded superior ROC-AUC and PR-AUC scores (Fig 3), validating their effectiveness. The details of the model's software and hardware environment as well as its hyperparameters can be found in S1 Table and S2 Table.

**Performance evaluation metrics.** The performance was assessed using ROC-AUC, PR-AUC. In ROC-AUC (TPR versus FPR for a series of threshold values), as shown Eq 1, the true-positive rate (TPR) and false-positive rate (FPR) are computed as:

$$TPR = \frac{TP}{TP + FN}, FPR = \frac{FP}{TN + FP} \tag{1}$$

Here TP denotes true positive; FN, false negative; TN, true negative; and FP, false positive. In PR-AUC (Eq 2), the precision and recall are computed by:

$$Precision = \frac{TP}{TP + FP}, Recall = \frac{TP}{TP + FN} \tag{2}$$

**Statistical analyses.** Performance benchmarking metrics, including ROC-AUC and PR-AUC, were computed using the Python package scikit-learn v.1.3.0. UMAP was conducted with the Python package umap-learn v.0.5.5. Sequence motifs were visualized using the Python package clogmaker v.0.8 with the "weblogo_protein" color scheme [51]. PyMOL v.2.5.8 was employed to visualize the 3D structure of TCR–pMHC complexes.

## Result

### THLANet outstanding performance

In this study, we conducted a rigorous and comprehensive evaluation of THLANet, pMTNet, PanPep, and TABR-BERT using the two benchmark datasets described in the Data Curation section. The compared models are detailed in Sect F of S1 Text. The performance was assessed using ROC-AUC) and PR-AUC, with TPR and FPR computed as described in Eqs 1 and 2 of the original manuscript. As shown in Fig 3a, we initially tested the models on the dataset containing 10x Genomics donor data, where THLANet outperformed the three baseline models. THLANet achieved median ROC-AUC and PR-AUC scores of 0.8906 and 0.8595,

respectively, compared to pMTNet's result of 0.8756 and 0.7873, PanPep's lower performance with scores of 0.8385 and 0.7259, and TABR-BERT's scores of 0.8686 and 0.8405. We trained the models on a dataset derived from VDJdb, IEDB, and McPAS-TCR, where THLANet consistently outperformed other baseline models (Fig 3b). Notably, THLANet achieved the most remarkable performance in the PR-AUC metric, attaining a score of 0.8503, indicating its superior capability in identifying positive samples compared to other baseline models. Detailed test data could be found in S3 Table.

We evaluated the PR-AUC of epitopes for representative TCRs in the test dataset. THLANet emerged as the best predictor for 12 of the 19 epitope targets (Fig 4c), including the EBV-BRLF1 epitope YVLDHLIVV (PR-AUC: 0.9899), the yellow fever virus peptide LLWNGPMAV (PR-AUC: 0.9145), and the hepatitis C virus peptide ATDALMTGY (PR-AUC: 0.9722).

This study thoroughly evaluated THLANet, pMTNet, PanPep, and TABR-BERT, confirming THLANet's superior performance in predicting TCR-pHLA binding. THLANet outperformed other baseline models on both the primary dataset, including 10x Genomics donor data, and the baseline dataset from VDJdb, IEDB, and McPAS-TCR, particularly excelling in the PR-AUC metric. This underscores its robust ability to identify positive samples, making it highly suitable for long-tail distributed datasets, a common feature of TCR-pHLA data. These findings validate THLANet's robustness across diverse immunological contexts and establish a strong foundation for its application in personalized immunotherapy, highlighting the significant potential of pretrained models in advancing precision medicine.

## THLANet exhibits potential in multi-cancer clinical applications

The interaction between TCRs and neoantigens is crucial in driving the progress of tumor immunotherapy. To assess whether THLANet can efficiently identify and predict neoantigens across various cancer types in clinical settings and validate its potential as a knowledge discovery tool, we characterized TCR-pHLA (T-cell receptor–human leukocyte antigen) interactions in several highly immunogenic tumor types.

We analyzed data from a patient with T-cell prolymphocytic leukemia (T-PLL) provided by 10X Genomics. This dataset utilizes feature barcode technology to generate single-cell 5' libraries and V(D)J-enriched libraries for TCR sequencing, along with highly multiplexed pHLA multimer reagents to determine binding specificity. The dataset includes 6,365 TCR clonotypes responsive to T-PLL, with samples having unique molecular identifiers (UMIs) ≥ 10 classified as positive and those with UMIs < 10 classified as negative (S1 Data). Furthermore, we incorporated peer-reviewed data from melanoma and gastrointestinal cancers, including 54 pairs specifically linked to melanoma and gastrointestinal cancers [44,45] (S3 Data and S4 Data).

Using THLANet, we predicted TCR-pHLA interactions in clinical cancer samples and compared its performance against three baseline models. In predictions for melanoma and gastrointestinal cancer samples, as expected, THLANet outperformed pMTNet, PanPep, and TABR-BERT, achieving median ROC-AUC and PR-AUC scores of 0.8375 and 0.7236, respectively (compared to pMTNet's median ROC-AUC and PR-AUC scores of 0.625 and 0.3990, respectively; PanPep's 0.5600 and 0.3416; and TABR-BERT's 0.6272 and 0.5625) (Fig 3c).

Similarly, in predictions for T-PLL samples, THLANet achieved superior performance, with a median ROC-AUC of 0.7169 and a median PR-AUC of 0.7066 (compared to pMTNet's median ROC-AUC of 0.5510 and PR-AUC of 0.5004; PanPep's median ROC-AUC of 0.6161 and PR-AUC of 0.5511; and TARB-BERT's median ROC-AUC of 0.5687 and PR-AUC of 0.4807) (Fig 3d).

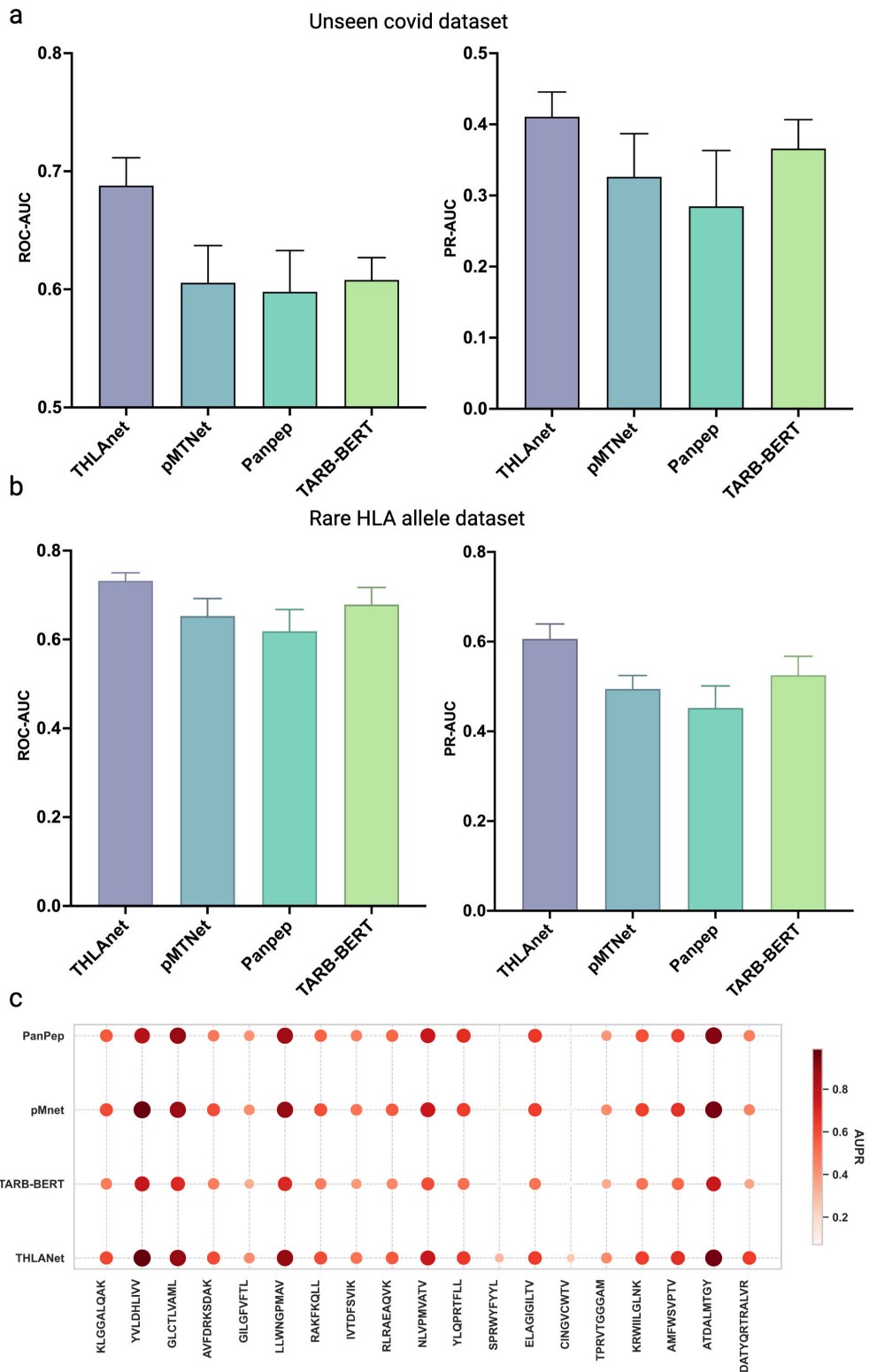

**Fig 4. (a) ROC-AUC and PR-AUC for THLANet, PanPep, pMTnet, and TABR-BERT on the unseen COVID-19 dataset from ImmuneCODE (2876 TCR-pHLA pairs).** (**b**) ROC-AUC and PR-AUC for THLANet and baseline models on low-frequency HLA class I dataset. (**c**) Comparison of area under the precision-recall curve values derived from PanPep, pMNet, TABR-BERT, and THLANet for 19 epitopes with more than ten binding T-cell receptors (TCRs) in the test dataset. A darker color and larger point size indicate a higher PR-AUC.

These findings underscore THLANet's ability to predict peptide-specific TCR binding interactions across diverse cancer types effectively, highlighting its significant potential for advancing exogenous TCR binding studies.

In conclusion, our study confirmed THLANet's effectiveness in identifying neoantigens that trigger CD8+ T-cell responses, underscoring its potential to advance immunotherapies, including ACT, TCR-T, and CAR-T strategies.

## Superior performance of THLANet on unseen datasets

To evaluate THLANet's robustness against data dependency, we tested its performance on a novel dataset from the ImmuneCODE project. This "unseen COVID-19 dataset" was curated to exclude any epitopes or HLA alleles present in THLANet's training data, ensuring a stringent test of generalization, 2876 unique TCR-pHLA pairs were collected. Without retraining, THLANet achieved a ROC-AUC of 0.6873 and a PR-AUC of 0.4107, significantly outperforming baseline models: PanPep (ROC-AUC: 0.5987, PR-AUC: 0.2848), pMTNet (ROC-AUC: 0.6056, PR-AUC: 0.3262), and TABR-BERT (ROC-AUC: 0.6077, PR-AUC: 0.3669) (Fig 4a). This robust generalization to unseen data is attributed to THLANet's hybrid architecture, integrating ESM-2 embeddings with a bilinear attention network to capture universal TCR-pHLA interaction patterns. Additionally, a sensitivity analysis with training data reduced to 40% of its original size showed minimal performance decline (ROC-AUC: 0.65, PR-AUC: 0.39), further validating THLANet's resilience (as shown in S5 Fig). These results underscore THLANet's exceptional capability to predict TCR-neoantigen interactions in diverse clinical scenarios, enhancing its potential for personalized immunotherapy applications.

## Evaluation of THLANet on low-frequency HLA class I alleles

To evaluate THLANet's generalizability to low-frequency HLA class I alleles, we curated a dataset of laboratory-validated TCR-pHLA interactions involving HLA-B*27:05, HLA-B*15:01, and HLA-A*24:02 from peer-reviewed studies [52–54]. Through random mismatching, we generated a dataset comprising 343 samples. Sequences were processed using ESM-2 and Transformer-Encoder modules, with features integrated via a bilinear attention network, consistent with our primary pipeline. As shown in Fig 4b, THLANet achieved a mean ROC-AUC of 0.7320 and a mean PR-AUC of 0.6062, demonstrating robust performance despite the underrepresentation of these alleles in the training data. Compared to baseline models (pMTNet: ROC-AUC 0.6527, PR-AUC 0.4943; PanPep: ROC-AUC 0.6184, PR-AUC 0.4521; TABR-BERT: ROC-AUC 0.6785, PR-AUC 0.5252), THLANet exhibited superior performance across all metrics. These findings underscore THLANet's clinical utility in immunotherapy, particularly for diverse patient cohorts, and highlight the need for broader allele representation in future datasets.

## Ablation experiment

We investigated the impact of the two primary modules in the network architecture on the performance of THLANet, focusing on the transformer encoder and ESM-2 architectures. To conduct this analysis, ablation studies were performed on each architecture individually, followed by a comparative evaluation using the validation dataset and two peer-reviewed cancer-related TCR-pHLA datasets (Fig 3e). The study revealed that both the transformer encoder and ESM-2 architectures moderately improved. While the ESM-2 architecture demonstrated exceptional ability in identifying positive samples, it showed less stability in performance

compared to the transformer encoder. By integrating both architectures, THLANet was able to leverage their respective strengths, achieving significant advancements in both predictive performance and stability.

We retrained and evaluated the model under the condition that only the Transformer encoder was retained as the encoding module. We observed the following results on the test dataset: AUC-ROC increased by 0.00173, and PR-AUC increased by 0.0373. While the improvement in AUC-ROC was not statistically significant at the 0.05 significance level (two-tailed Wilcoxon signed-rank test), the increase in PR-AUC was significant ($p < 0.05$). On two independent cancer-related datasets, the transformer encoder achieved improvements of AUC-ROC = 0.1302 ($p < 0.05$) and PR-AUC = 0.1444 ($p < 0.05$), as well as AUC-ROC = 0.1394 ($p < 0.05$) and PR-AUC = 0.0309 ($p < 0.05$).

Under the ablation of the Transformer encoder module, we retrained and evaluated the model. Leveraging the pretrained protein parameters of the ESM-2 model, we embedded protein sequences to capture richer encoding features. Following ablation of the ESM-2 architecture, we retrained and reevaluated the model. On the test dataset, we observed AUC-ROC increased by 0.00804, and AUC-ROC increase of 0.0139 ($p < 0.05$). On two independent cancer-related datasets, the ESM-2 architecture achieved improvements of AUC-ROC = 0.2038, PR-AUC = 0.1830, and AUC-ROC = 0.0258 ($p < 0.005$), PR-AUC = 0.0309 ($p < 0.005$).

According to the findings of this experiment, the integration of the ESM-2 module and the Transformer-Encoder module provides richer sequence encoding features for THLANet, while also enhancing the overall generalization capability of the model, enabling more accurate predictions of clinical sample data.

## THLANet provides explanation for the spatial structure of the TCR-pHLA interaction

To investigate the binding mechanisms of TCR-pHLA interactions in the microenvironment of CD8+ T-cells, we performed computational simulations of mutation analyses to identify significant changes in TCR and pHLA binding caused by mutations in CDR3 residues. To support our experiments and validate our findings in 3D space, we compiled 112 samples from the IEDB containing 3D structures of TCR-pHLA complexes, the data filtering criteria are provided in Sect D of S1 Text. The alanine scanning technique used in biophysical studies provided the conceptual framework for this analysis. Using computational simulations of alanine scanning, we performed base mutations on all TCRs in the 112 test groups, recording the predicted differences between the wild-type TCRs and their mutated counterparts. Alanine scanning was chosen for TCR-pHLA binding analysis due to its distinct advantages in protein interaction studies. Alanine's methyl side chain, small and non-polar, minimizes perturbation to the protein backbone conformation, avoiding steric or chemical interference from charged or bulky side chains (e.g., lysine). Compared to other amino acids, alanine precisely evaluates residue contributions to binding and, being common in natural proteins, reduces non-specific effects. To investigate the role of the complementarity-determining region 3 (CDR3) in TCR–pHLA binding, CDR3 sequences were evenly divided into five segments (N-terminal to C-terminal as shown Fig 5a). A sliding window approach was employed, dynamically adjusting segment lengths based on total sequence length (10–25 residues) to approximate equal divisions, with the central segment prioritized to cover key contact regions. For sequences with lengths not divisible by five, remaining residues were evenly distributed to adjacent segments. This segmentation strategy was validated against contact points from PDB crystal structures, ensuring structural consistency. Segmented CDR3 data underwent alanine

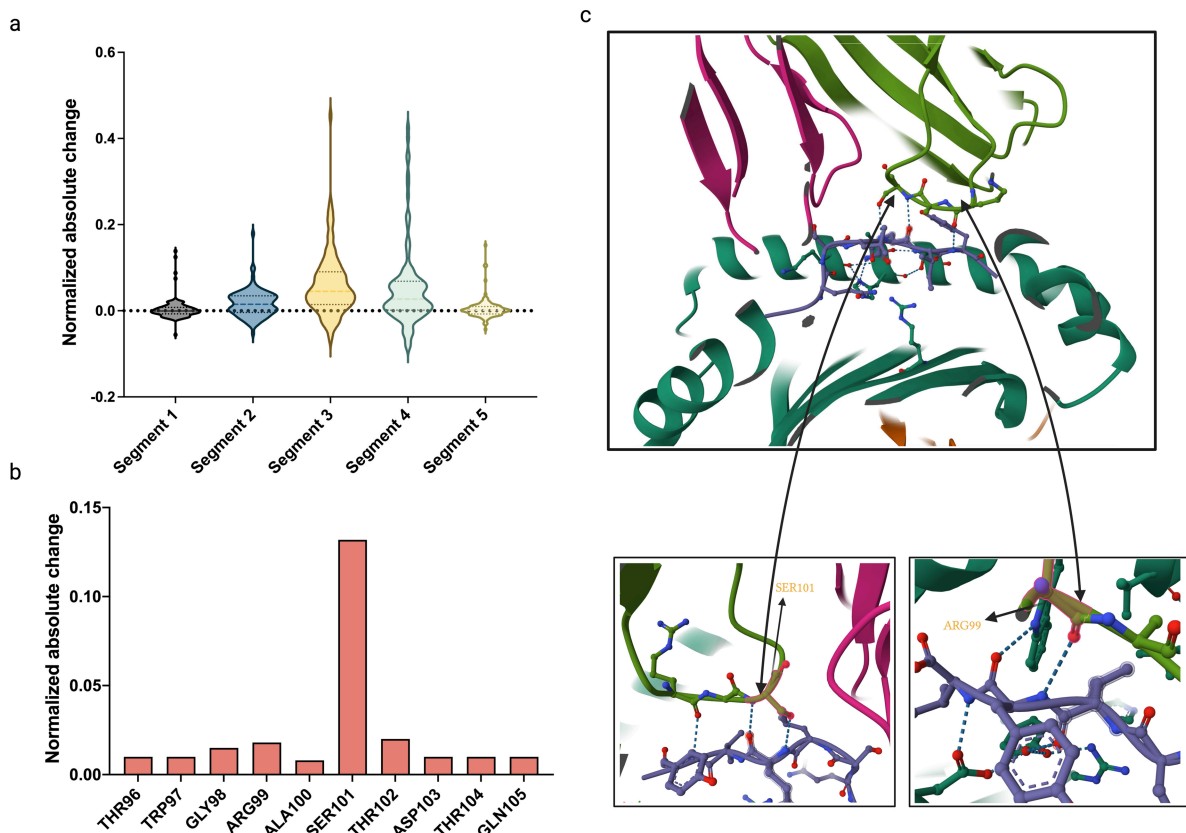

**Fig 5. Validation of the T-cell receptor–human leukocyte antigen (TCR-pHLA) network (THLANet) in identifying critical sites within the three-dimensional (3D) crystal structure. (a)** The complementarity-determining region 3 (CDR3) residues in the middle of the TCR sequence exhibited significant changes in scores predicted by THLANet, as validated through alanine scanning mutagenesis. The CDR3 sequence was divided into five equal-length segments for the alanine scanning analysis. **(b-c)** Predicted score changes for amino acid residues in the CDR3 of an example TCR–peptide-major histocompatibility complex TCR-pMHC structure (PDB ID: 8GON). The 3D structure of 8GON is illustrated: green, CDR3 of the TCRβ chain; magenta, TCRα chain; tints, other regions of the TCRβ chain; violet, antigen.

scanning, and the resulting features were input into THLANet for binding prediction. Predicted binding scores were statistically analyzed based on the position of alanine-substituted residues within the segments. In Fig 5a–5c, we present the TCR-pHLA structure generated by X-ray diffraction (PDB ID: 8GON, Resolution: 2.60 Å). Following computational simulations of alanine scanning, residue-by-residue predictions were conducted (Fig 5b). The residues SER101 and ARG99 exhibited the most significant predictive differences, attributed to their central locations in CDR3, where they made the most contact with pHLA (Fig 5c). Notably, the SER101 residue simultaneously contacts two antigen residues, resulting in even greater predictive changes. In summary, our study demonstrates that, although THLANet was developed at the one-dimensional sequence level, it exhibits the capacity to elucidate the intrinsic properties of peptide-TCR interactions and to identify critical sites within three-dimensional structures.

The importance of the CDR3 central region in TCR-pHLA interactions is well-recognized [18]. However, THLANet provides novel quantitative evidence through computational alanine scanning, precisely identifying key residues (e.g., SER101, ARG99) impacting binding affinity (Fig 5a). Analysis of 112 high-quality TCR-pHLA 3D structures revealed spatial

interaction patterns, offering new insights into binding mechanisms, complementing existing knowledge, and supporting immunotherapy applications.

## Discussion

In this study, we present THLANet, an innovative deep learning framework designed to accurately predict TCR-neoantigen binding specificity. Identifying TCRs that recognize specific neoantigens is pivotal for personalized immunotherapy, as not all neoantigens elicit T-cell responses. Our results underscore the critical role of precise TCR-pHLA interaction prediction in enhancing cancer immunotherapy. THLANet surpasses existing models across multiple evaluation metrics, demonstrating substantial promise for clinical applications.

THLANet introduces methodological advancements that set it apart from state-of-the-art models such as PanPep, pMTnet, and TABR-BERT. Unlike PanPep, which relies on meta-learning and neural Turing machines for task adaptation, THLANet integrates ESM-2 embeddings, a bilinear attention mechanism, and sinusoidal positional encoding within a Transformer-Encoder architecture. This hybrid design effectively captures intricate TCR-pHLA interactions by combining sequence and structural features, achieving superior prediction accuracy across diverse datasets. In contrast to pMTnet's stacked autoencoders and LSTM-based sequential encoding, THLANet's bilinear attention mechanism ensures robust integration of TCR and pHLA embeddings, minimizing information loss and enhancing generalization to novel epitopes. Compared to TABR-BERT's separate TCR and pMHC encoding modules, THLANet's unified encoding approach reduces computational complexity while maintaining high precision. Additionally, THLANet's segment-based CDR3 analysis, validated against PDB crystal structures, divides sequences into five segments to provide interpretable insights into key binding residues, directly supporting neoantigen vaccine design and TCR-T therapy optimization.

In conclusion, THLANet marks a significant leap forward in T-cell-mediated immunotherapy, paving the way for more effective cancer treatments. Future research directions include: (1) extending THLANet to predict TCR binding with HLA class II-presented antigens, and (2) integrating THLANet with genomic technologies to validate predicted neoantigens across diverse cancer types and patient cohorts, further advancing personalized immunotherapy.

## Supporting information

**S1 Text. Contains supplemental text, methods.**
(DOCX)

**S1 Fig. (a) Definition of the HLA pseudo sequence. (b) BLOSUM62 matrix.**
(TIFF)

**S2 Fig. Sequence motifs of CDR3$\beta$ representing the epitope-specific TCRs for protein epitopes.**
(TIFF)

**S3 Fig. The t-SNE plots of the data from four donors in the 10X Genomics dataset.**
(TIFF)

**S4 Fig. The training model structure of THLANet.**
(TIFF)

**S5 Fig. The model's sensitivity analysis with respect to the training dataset.**
(TIFF)

**S1 Table. Software and Hardware Environment.**
(XLSX)

**S2 Table. Model parameter configuration.**
(XLSX)

**S3 Table. The ROC-AUC and PR-AUC results of THLANet and baseline models.**
(XLSX)

**S4 Table. The dataset used for the comparative models in this manuscript.**
(XLSX)

**S5 Table. The information of peptides with known binding TCR$\beta$(CDR1, CDR2, CDR3) sequences.**
(XLSX)

**S6 Table. The information of peptides with known binding TCR$\beta$ sequences.**
(XLSX)

**S7 Table. The information of peptides with known binding TCR($\alpha+\beta$) sequences.**
(XLSX)

**S1 Data. Prolymphocytic leukemia dataset.**
(XLSX)

**S2 Data. Data curation of the 3D TCR-Antigen-HLA complex from PDB.**
(XLSX)

**S3 Data. Human Gastrointestinal Cancer-Associated TCR-pHLA Dataset.**
(XLSX)

**S4 Data. Melanoma-Associated TCR-pHLA Dataset.**
(XLSX)

**S5 Data. The list of pMHC multimers used by 10x Genomics Dataset.**
(XLSX)

## Author contributions

**Conceptualization:** Xu Long, Xiaokun Li.

**Data curation:** Xu Long, Qiang Yang.

**Formal analysis:** Weihe Dong, Suyu Dong.

**Funding acquisition:** Xiaokun Li, Kuanquan Wang, Gongning Luo.

**Investigation:** Xu Long, Qiang Yang.

**Methodology:** Xu Long.

**Project administration:** Xu Long, Xin Gao, Guohua Wang.

**Supervision:** Xiaokun Li, Xin Gao, Guohua Wang.

**Validation:** Xu Long.

**Writing – original draft:** Xu Long.

**Writing – review & editing:** Xianyu Zhang, Tiansong Yang.

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
