## [Decision Letter · Decision Letter 0]

12 Jun 2025

PCOMPBIOL-D-25-00736

THLANet: A Deep Learning Framework for Predicting TCR-pHLA Binding in Immunotherapy Applications

PLOS Computational Biology

Dear Dr. Li,

Thank you for submitting your manuscript to PLOS Computational Biology. After careful consideration, we feel that it has merit but does not fully meet PLOS Computational Biology's publication criteria as it currently stands. Therefore, we invite you to submit a revised version of the manuscript that addresses the points raised during the review process.

Please submit your revised manuscript within 60 days Aug 12 2025 11:59PM. If you will need more time than this to complete your revisions, please reply to this message or contact the journal office at ploscompbiol@plos.org. Please include the following items when submitting your revised manuscript:

We look forward to receiving your revised manuscript.

Kind regards,

Chaok Seok

Academic Editor

PLOS Computational Biology

Dominik Wodarz

Section Editor

PLOS Computational Biology

**Journal Requirements:**

At this stage, the following Authors/Authors require contributions: Qiang Yang, Weihe Dong, Kuanquan Wang, Xiaokun Li, Suyu Dong, Gongning Luo, Xianyu Zhang, Tiansong Yang, Xin Gao, Guohua Wang, and Xu Long. Please ensure that the full contributions of each author are acknowledged in the "Add/Edit/Remove Authors" section of our submission form.

3) Please ensure that all Figure files have corresponding citations and legends within the manuscript. Currently, Figure 2 in your submission file inventory does not have an in-text citation. Please include the in-text citation of the figure.

Potential Copyright Issues:

i) Figure 1a. Please confirm whether you drew the images / clip-art within the figure panels by hand. If you did not draw the images, please provide (a) a link to the source of the images or icons and their license / terms of use; or (b) written permission from the copyright holder to publish the images or icons under our CC BY 4.0 license. Alternatively, you may replace the images with open source alternatives. See these open source resources you may use to replace images / clip-art:

3) If any authors received a salary from any of your funders, please state which authors and which funders.

6) Please ensure that the funders and grant numbers match between the Financial Disclosure field and the Funding Information tab in your submission form. Note that the funders must be provided in the same order in both places as well. Currently, "the King Abdullah University of Science and Technology (KAUST) Office of Research Administration (ORA) under Award No REI/1/5234-01-01, REI/1/5414-01-01, REI/1/5289-01-01, REI/1/5404-01-01, REI/1/5992-01-01, URF/1/4663-01-01, Center of Excellence for Smart Health (KCSH), under award number 5932, Center of Excellence on Generative AI, under award number 5940" are missing from the Funding Information tab.

7) Please provide a completed 'Competing Interests' statement, including any COIs declared by your co-authors. If you have no competing interests to declare, please state "The authors have declared that no competing interests exist". 

**Reviewers' comments:**

Reviewer's Responses to Questions

Reviewer #1: The paper introduces a comprehensive and meaningful deep learning framework for predicting the interactions between T-cell receptors (TCRs) and antigens presented by Class I human leukocyte antigens (HLA-I), a task of critical importance in the development of cancer immunotherapy and immunological theoretical research. By integrating diverse clinical data and advanced deep learning techniques, THLAnet significantly outperforms existing tools in comparative evaluations. Notably, through the analysis of complementarity-determining region 3 (CDR3) sequences and the application of computer-simulated alanine scanning techniques, this study elucidates the 3D binding conformations of TCRs with antigens, offering novel insights into TCR-antigen interactions.

There are some suggestions for this manuscript.

(1) Considering the complexity of TCR-pHLA interactions and their immunogenicity, what preprocessing steps were applied to the data before training the model? How were the features selected or designed to effectively capture the nuances of these interactions?

(2) Could you provide more details on the experimental setup, including the hardware and software environment? How does the study ensure the reproducibility of its results by the broader scientific community?

(3) TCRs are very promiscuous, with a single TCR capable of recognizing more than a million peptides. Do you think this has an impact on the false negatives in the TCR data? What impact does this have on training? Please add some comments to the discussion.

(4) Could you define better what you mean by sinusoidal encoding and why you do it? Readers not familiar with Transformer may not understand.

(5) Please consider including your training data in the GitHub repo or providing an accessible link. It can help and guide the readers to repeat your work.

(6) Line 57: “We began by analyzing data generated using the 10x Genomics Chromium single-cell immunoassay platform.” Please provide a detailed description explaining how these data were obtained and processed.

(7) The manuscript mentions the use of alanine scanning to identify critical binding sites. Why was alanine chosen for simulated amino acid substitutions instead of other amino acids? This may be puzzling for researchers without a biology background; please include an explanation in the text.

(8) The study utilizes ESM-2, a protein language model. To my knowledge, ESM-2 comes in various parameter sizes—why was the 650M parameter version selected? What considerations guided your choice of protein language model parameters?

Reviewer #2: This Manuscript proposes a deep learning framework, THLANet, for predicting the binding specificity of T-cell receptors (TCRs) to neoantigens presented by class I human leukocyte antigens (HLAs). By integrating the Evolutionary Scale Model (ESM-2) and Transformer-Encoder modules, THLANet significantly outperforms existing models in sequence feature extraction and generalization capabilities. It demonstrates application potential in clinical data across various cancer types, such as melanoma, gastrointestinal cancers, and T-cell prolymphocytic leukemia. Through alanine scanning and 3D structural analysis, the article not only validates the model at the sequence level but also provides insights into the spatial mechanisms of TCR-pHLA binding, enhancing the biological significance of the results. The article exhibits significant scientific value in data integration, model architecture design, and clinical validation, particularly for its potential applications in personalized immunotherapies (e.g., ACT, TCR-T, and CAR-T). The article is well-structured and provides detailed supplementary materials for further reference. However, certain details require further clarification to enhance scientific rigor and readability.

Minor issues:

1. The article mentions the use of the Adam optimizer, a learning rate of 5×10⁻⁴, a batch size of 128, etc. (page 7), but does not explain how these hyperparameters were determined.

2. How did you split the data into training and validation? Did you ensure no overlap or closely similar sequences between TCRs or antigen sequences?

3. How did you get the test dataset in Section 3.1?

4. Where did the “peer-reviewed data from melanoma and gastrointestinal cancers” come from? Please cite the related references.

5. The conclusion of Section 3.4 that the central part of CDR3 exhibited higher importance in TCR-pHLA interactions is a well-known notion.

6. It would be better to add a figure to depict the last MLP module of Figure 2b.

7. The input to ESM-2 and to Transformer-Encoder model were not described explicitly, and some details need to be added.

Reviewer #3: This manuscript presents THLANet, a deep learning model designed to predict whether a T-cell receptor (TCR) will bind a peptide presented by a class I HLA molecule (pHLA). The authors combine a large pretrained protein language model (ESM-2) with a custom transformer encoder to encode TCR bata chain sequences (CDR3 region), peptide antigens, and HLA information. A bilinear attention fusion is used to integrate features from these two encoders, and the fused representation is fed into a convolutional neural network and multi-layer perceptron classifier. The model is trained on a compiled dataset of TCR-pHLA pairs drawn from single-cell pHLA multimer experiments and public databases. The authors report that it outperforms other methods. While this study addresses an important challenge in immunoinformatics and introduces an innovative modelling approach, I find that the manuscript in its current form requires major revisions before it can be considered for publication.

Major Comments

1) The choice of ESM-2 as a general protein embedding tool raises concerns about its appropriateness for describing TCR–pMHC interactions. Given that ESM-2 is not specialised for immune recognition, the manuscript should explicitly demonstrate that ESM-2 embeddings effectively separate or distinguish TCR and pMHC sequences in embedding space. Alternatively, developing or employing a language model specifically trained for TCR–pMHC data would strengthen confidence in the model's predictive capabilities.

2) The benchmarking of THLANet appears limited, relying mainly on the authors’ curated test dataset. I strongly suggest performing a comprehensive and rigorous benchmark evaluation using standard public datasets (VDJdb and McPAS-TCR) separately, clearly distinguishing training and testing data to demonstrate broader robustness and generalisability.

3) Although data dependency is identified by the authors as a limitation in other methods, the current manuscript lacks explicit testing of THLANet’s own potential data dependency issues. Additional experiments, such as systematically varying training set sizes, or evaluating performance on completely unseen epitopes or HLAs, are necessary to conclusively address how THLANet handles data dependency.

4) The clinical relevance of peptides used for training and evaluation is unclear. The manuscript should clearly state their intended role and explicitly acknowledge whether these peptides have direct clinical relevance or serve only as proxies to validate binding prediction capability.

5) Typically, neoantigen discovery pipelines begin with the filtering of peptides based on their ability to bind MHC molecules. The manuscript does not clarify if peptides included in the study are experimentally validated MHC binders or predicted binders. Clarifying this point explicitly would address potential concerns about the biological validity of negative samples and their potential impact on model performance.

6) Regarding novelty, the manuscript utilises common deep-learning building blocks such as the ESM-2 language model and transformer architectures, making the novelty somewhat incremental. To justify claims of innovation, the manuscript should clearly articulate what distinct methodological contributions or novel applications THLANet provides compared to existing state-of-the-art approaches.

7) The dataset shows significant biases towards a limited set of HLA alleles (mostly HLA-A), with underrepresentation of other HLA class I alleles. This raises questions about THLANet’s general applicability. The manuscript needs to demonstrate explicitly whether the model’s performance remains consistent across diverse HLA class I alleles and discuss the implications of potential performance drops for rare alleles.

8) The manuscript claims to "unveil 3D binding conformations" of TCR–pHLA complexes, but the evidence provided relies solely on in silico mutational analysis. While computational alanine-scanning experiments are a useful addition, this claim should be toned down to avoid overstating the findings. The authors essentially identify predicted critical residues in TCR and antigen interactions, correlating these predictions with known structural contacts in limited examples. This analysis provides predictive structural insights but does not determine or "unveil" actual 3D conformations. The authors should rephrase such statements to clearly reflect that these are predictive insights. Additionally, more explicit explanation of the method used for dividing CDR3 sequences into segments (beyond figure captions) is required in the main text.

Minor comments:

1) I found some sentences are extremely difficult to follow (For example, lines 6-8). The manuscript would benefit from significant editing for clarity.

2) No IRB for the blood collection from the healthy donors?

3) No code availability?

4) No funding information?

**Have the authors made all data and (if applicable) computational code underlying the findings in their manuscript fully available?**

Reviewer #1: **No: **

Reviewer #2: None

Reviewer #3: **No: **I cannot find code or repository information in the manuscript.

PLOS authors have the option to publish the peer review history of their article (what does this mean?). If published, this will include your full peer review and any attached files.

Reviewer #1: No

Reviewer #2: No

Reviewer #3: No

**Figure resubmission:**
---

## [Decision Letter · Decision Letter 1]

28 Aug 2025

Dear Prof Li,

We are pleased to inform you that your manuscript 'THLANet: A Deep Learning Framework for Predicting TCR-pHLA Binding in Immunotherapy Applications' has been provisionally accepted for publication in PLOS Computational Biology.

Best regards,

Chaok Seok

Academic Editor

PLOS Computational Biology

Dominik Wodarz

Section Editor

PLOS Computational Biology

Reviewer's Responses to Questions

**Comments to the Authors:**

Reviewer #1: The author has thoroughly and carefully revised the manuscript in response to my questions, resolving all my concerns.

Reviewer #2: It is revised well

Reviewer #3: The authors have thoroughly addressed every comment, making substantial improvements to both the content and clarity of the manuscript. All the major comments have been fully resolved through additional experiments, analyses, and textual revisions, and the four minor comments have been adequately fixed through clarifications and added information.

Notably, the authors performed extensive new work (fine-tuning the language model, benchmarking on public datasets, testing unseen data and rare HLAs) to bolster their claims. They also clarified methodological details and toned down any overstatements. These changes greatly strengthen the manuscript’s validity, transparency, and significance.

**Have the authors made all data and (if applicable) computational code underlying the findings in their manuscript fully available?**

Reviewer #1: Yes

Reviewer #2: None

Reviewer #3: Yes

PLOS authors have the option to publish the peer review history of their article (what does this mean?). If published, this will include your full peer review and any attached files.

Reviewer #1: No

Reviewer #2: No

Reviewer #3: No

---

## [Editor Report · Acceptance letter]

PCOMPBIOL-D-25-00736R1

THLANet: A Deep Learning Framework for Predicting TCR-pHLA Binding in Immunotherapy Applications

Dear Dr Li,

I am pleased to inform you that your manuscript has been formally accepted for publication in PLOS Computational Biology. Your manuscript is now with our production department and you will be notified of the publication date in due course.

With kind regards,

Zsofia Freund
